# Neural Correlates of Letter and Semantic Fluency in Primary Progressive Aphasia

**DOI:** 10.3390/brainsci12010001

**Published:** 2021-12-21

**Authors:** Marianna Riello, Constantine E. Frangakis, Bronte Ficek, Kimberly T. Webster, John E. Desmond, Andreia V. Faria, Argye E. Hillis, Kyrana Tsapkini

**Affiliations:** 1Department of Neurology, Johns Hopkins School of Medicine, Baltimore, MD 21205, USA; marianna.riello@gmail.com (M.R.); bficek1@jhmi.edu (B.F.); Kwebste1@jhmi.edu (K.T.W.); jdesmon2@jhmi.edu (J.E.D.); argye@jhmi.edu (A.E.H.); 2Department of Biostatistics, Johns Hopkins School of Public Health, Baltimore, MD 21227, USA; cfranga1@jhu.edu; 3Department of Radiology, Johns Hopkins School of Medicine, Baltimore, MD 21227, USA; afaria1@jhmi.edu; 4Department of Psychiatry and Behavioral Sciences, Johns Hopkins School of Medicine, Baltimore, MD 21227, USA; 5Department of Otolaryngology, Head and Neck Surgery, Johns Hopkins School of Medicine, Baltimore, MD 21227, USA; 6Department of Cognitive Science, Johns Hopkins University, Baltimore, MD 21218, USA; 7Department of Physical Medicine and Rehabilitation, Johns Hopkins School of Medicine, Baltimore, MD 21205, USA

**Keywords:** primary progressive aphasia, grey matter volumes, phonological fluency, letter fluency, category fluency, semantic fluency

## Abstract

Verbal fluency (VF) is an informative cognitive task. Lesion and functional imaging studies implicate distinct cerebral areas that support letter versus semantic fluency and the understanding of neural and cognitive mechanisms underlying task performance. Most lesion studies include chronic stroke patients. People with primary progressive aphasia (PPA) provide complementary evidence for lesion-deficit associations, as different brain areas are affected in stroke versus PPA. In the present study we sought to determine imaging, clinical and demographic correlates of VF in PPA. Thirty-five patients with PPA underwent an assessment with letter and category VF tasks, evaluation of clinical features and an MRI scan for volumetric analysis. We used stepwise regression models to determine which brain areas are associated with VF performance while acknowledging the independent contribution of clinical and demographic factors. Letter fluency was predominantly associated with language severity (R^2^ = 38%), and correlated with the volume of the left superior temporal regions (R^2^ = 12%) and the right dorsolateral prefrontal area (R^2^ = 5%). Semantic fluency was predominantly associated with dementia severity (R^2^ = 47%) and correlated with the volume of the left inferior temporal gyrus (R^2^ = 7%). No other variables were significantly associated with performance in the two VF tasks. We concluded that, independently of disease severity, letter fluency is significantly associated with the volume of frontal and temporal areas whereas semantic fluency is associated mainly with the volume of temporal areas. Furthermore, our findings indicated that clinical severity plays a critical role in explaining VF performance in PPA, compared to the other clinical and demographic factors.

## 1. Introduction

Verbal fluency tasks (letter and semantic fluency) are the most common neuropsychological tests used to assess verbal functioning [1]. They are mentioned in most publications on verbal deficits, more often than any other task except, perhaps, naming. In the verbal fluency tasks participants are asked to produce as many words as possible starting with a given letter (letter fluency task, also called letter fluency, phonemic/phonological and/or word fluency) or words within a specific semantic category (semantic or category fluency task). Both verbal fluency tasks might rely on similar executive cognitive skills, such as initiation (the ability to use attention to generate the word), self-monitoring (suppressing the activation of inappropriate responses—e.g., semantically related words, or repetition), cognitive flexibility (ability to rapidly switch strategies) [2], and other cognitive functions related to memory and language (available semantic and lexical knowledge from which to identify relevant items) and recall (ability to retrieve items from verbal declarative memory) [3]. Additionally, two important components were recognized among both tasks: “clustering”, the number of items in each cluster within letter or semantic subcategories, and “switching”, the number of switches between subcategories. These two components are shared by both tasks, with the switching component being more related to the frontal lobe functioning [4].

With regard to brain correlates of letter and semantic fluency, studies have demonstrated that letter fluency was largely associated with the integrity of executive functions (i.e., strategic searches in the phonological lexicon), thus being more dependent upon the frontal lobe [5,6,7]. By contrast, semantic fluency has been associated with the integrity of the temporal lobes (i.e., strategic searches in the semantic system). It should be noted that other areas were also found to be significantly associated with deficits in both letter and semantic fluency, such as the parietal cortex [7]. Additionally, a lateralized specialization has been reported, since some studies have shown reduced letter fluency following left rather than right frontal lesions [3,7,8,9], while other studies have reported involvement of right lesions in both verbal fluency tasks [10,11]. A meta-analysis of patients with focal lesions reported large and comparable effects of frontal lesions for both tasks, in contrast to a larger involvement of temporal regions for the semantic compared to phonemic fluency deficit [3].

A word of caution on the notion of fluency. Several studies use the term “fluency” to refer to spontaneous speech fluency tasks in PPA using picture description, story-telling, or analysis of speech [12,13,14,15,16,17]. These studies mostly investigated the rate of speech (number of words per minute), sentence length, initiation of speech (effortful vs. automatic), word choice (substantive vs. relational), pauses, perseveration, paraphasias, and prosody or pronunciation. In fact, picture description tasks provide a wealth of information for classification of PPA as shown in these studies. We have recently also shown their value using an automated, end-to-end machine learning method that reliably classified patients into the three main PPA variants with 80% accuracy, similar to imaging and neuropsychological classifications [18,19]. In the present study, however, when we refer to verbal fluency (VF) we are referring only to letter and semantic category (semantic) fluency tasks as specified by [1] (see [20], for an in-depth discussion of the notion of fluency and its measurement in aphasia).

Several studies have looked at fluency measures in progressive language disorders such as primary progressive aphasia (PPA). PPA is associated with variable patterns of degeneration, especially in the left hemisphere, that result in great variability of language deficits [21,22]. PPA is defined by an isolated and gradual dissolution of language processing, starting with anomia and later progressive loss of fluency [22]. It is characterized by three variants: a nonfluent variant (nfvPPA), a logopenic variant (lvPPA), and a semantic variant (svPPA) in most recent classifications, based on language/cognitive, clinical, and neuro-anatomical characteristics [21]. Patients with nfvPPA can present with abnormality of grammar in spoken or written language, and/or apraxia of speech, in the presence of relatively preserved single-word comprehension and present with rather left frontal atrophy patterns [21,22]. Patients with svPPA (previously known as semantic dementia) show abnormal single-word comprehension and naming due to loss of word knowledge, despite relatively preserved grammar and fluency, and present with temporal (left dominant but bilateral) atrophy [21]. These patients typically maintain fluent but empty language output, or use words of high familiarity, as we recently showed [23], and can present with a reverse concreteness effect [24,25]. Finally, patients with lvPPA are characterized by anomia, poor word retrieval in spontaneous speech, difficulty repeating sentences due to phonological short-term memory impairment that contributes to a dysfluent profile, and they present with inferior parietal/posterior temporal atrophy [21].

Behavioral studies in verbal fluency have shown that patients with PPA, and those with language deficits in other cognitive and speech neurodegenerative disorders (e.g., mild cognitive impairment and primary progressive apraxia of speech, respectively), produce fewer words in both letter and semantic fluency tasks than healthy subjects [23,24,26,27,28,29,30,31]. Semantic fluency was severely impaired in semantic dementia and was associated with temporal lobe hypometabolism [24,32] while letter fluency was more notably impaired in nfvPPA [24], although both letter and semantic fluency were impaired [32]. Subsequent studies demonstrated that letter fluency was more impaired than semantic fluency in what may correspond to a combination of nfv and lvPPA [30] in today’s classification terms [21]. In semantic fluency, nfv/lvPPA and svPPA had similar performance, but in letter fluency, the performance of participants with nfv/lvPPA was significantly worse than that of participants with svPPA [30]. However, when the frequencies of the words generated in semantic fluency (animal category) were compared to each other, the nfv/lvPPA group produced significantly less frequent animals than the svPPA group. Interestingly, we replicated the same effect in our recent study using a machine learning approach to detect differences between variants in several word property measures (e.g., familiarity, imageability). We found that svPPA produced significantly more familiar words (in both semantic and letter fluency) than both nfvPPA and lvPPA groups [23]. Finally, the only study that had lvPPA as a separate PPA variant [27] showed that they were equally impaired in letter and semantic fluency.

Despite the fact that several studies in PPA have reported on verbal fluency performance [26], there is only one study [29] that looked at the neural predictors of letter and semantic fluency in nfvPPA, svPPA, and behavioral/dysexecutive disorder (the latter frontotemporal dementia variant is beyond the scope of the discussion here). Libon and colleagues (2009) did not compare nfvPPA and svPPA in letter and semantic fluency, but rather the difference in fluency measures within each variant. Behaviorally, they showed that patients with svPPA were more impaired in semantic than letter fluency and patients with nfvPPA were equally impaired in both tasks. Employing a voxel-based morphometry (VBM) analysis, Libon and colleagues (2009) showed that, in svPPA, both letter and semantic fluency correlated with atrophy in the anterior and inferior left temporal regions, whereas in nfvPPA, letter fluency correlated with left temporal atrophy and semantic fluency correlated with right frontal atrophy. The latter results seem somewhat counterintuitive, especially in nfvPPA, since the epicenter of atrophy in this variant lies in the frontal cortices. The results may be explained by the great variability of the syndrome as shown in the above behavioral studies. Variability renders PPA a great syndrome model to ask questions of brain–language relationships [33], but also increases the Standard Error (SE) and therefore the ability to detect differences. Therefore, inferential statistics have reduced power to detect effects in a small sample size in each variant (N = 11 in nfvPPA and N = 10 in svPPA). Another possible reason for the discrepancy between the Libon et al. (2009) paper and the previous literature on areas responsible for letter and semantic fluency may be the degenerative nature of the disease. With progression, atrophy in neurodegenerative disorders extends beyond the epicenter and usually to adjacent or connected areas, and small changes in atrophy in these remote areas may have disproportionally detrimental effects in language and cognitive functions. For this reason, we found it useful to include an index of disease severity in our study.

Furthermore, demographic factors may play a crucial role in verbal fluency performance. Studies on normal populations have reported heterogeneous data on the role of demographic factors in verbal fluency: a gender effect (women outperform men) has been found on letter [34,35,36] and semantic fluency [37]. However, other studies have not confirmed a gender effect on either letter [38,39,40] or semantic fluency [41]. With regard to an age effect, some studies have found that normal aging affects letter [38] and semantic fluency [4,35,37,40,41], but others have not [34,42]. Instead, education has been found to be a significant predictor for both letter and semantic fluency [35,36,40]. The effect of clinical or demographic variables in determining the neural predictors of letter and semantic fluency in PPA has not yet been addressed, to the best of our knowledge. A previous study in PPA [43] demonstrated that women show significantly different performances and rates of decline on both verbal fluency tasks compared to men. Here we address the limitations of previous studies. Furthermore, we include lvPPA, with probable AD pathology [44] confirmed in approximately 70% of diagnosed patients], which has not yet been included in most previous verbal fluency analyses.

The aim of this study was to investigate the differences in letter and semantic fluency between the three main PPA variants, as well as the cerebral underpinnings of letter and semantic fluency, while also evaluating the role of other clinical or demographic factors, such as clinical severity or education. We have previously reported that these latter variables affect naming in PPA [45]. We included volume in regions of interest (selected according to the previous literature on verbal fluency in PPA), the clinical variables that may independently affect severity of language and cognitive symptoms in PPA (years post-onset, dementia severity, and language severity), and demographic factors (age, gender, and education), as predictors in regression models.

## 2. Methods

### 2.1. Participants

Thirty-five individuals with PPA (10 individuals with lvPPA, 17 individuals with nfvPPA, 8 individuals with svPPA) were included in the study. All participants (16F, age range: 51–82 years) met diagnostic criteria for PPA according to current consensus after clinical, imaging, language, and neuropsychological examination [21]. Participants were enrolled from the Johns Hopkins Outpatient Center’s PPA Clinic or Frontotemporal and Young-Onset Dementia Clinic or referred by physicians specializing in PPA to participate in a clinical trial (ClinicalTrials.gov, accessed on 15 December 2021 Identifier: NCT02606422). All were native, monolingual English speakers, with normal hearing and vision and no history of cerebrovascular accident, psychiatric deficits or other neurological disorders. Participants provided written informed consent for their participation in the study. The experimental procedures were approved by the ethical committee for experiments involving humans of Johns Hopkins Hospital Institutional Review Board (No: NA_00071337).

### 2.2. Materials and Procedures

Fluency abilities were tested with the letter fluency test [46] by asking participants to verbally generate as many words as possible beginning with the letters F, A, and S, allowing one minute for each letter, and with the semantic fluency test [47] by asking participants to verbally generate as many words as possible from the semantic categories of animals, fruits, and vegetables, allowing one minute for each category. The final raw score represented the sum of the words verbally pronounced in the time established for each category belonging to each task. Repetitions of the same word were not included in the final score. 

Participants were also assessed with frontotemporal lobar degeneration-modified clinical dementia rating scale (FTLD-CDR) [48]. The range of possible scores is 0 to 24, with higher scores indicating higher severity. The FTD-CDR score also includes an evaluation on eight subscales: memory, orientation, judgment and problem solving, community affairs, home and hobbies, personal care, behavior comportment, personality severity, and language. For the regression analyses, we used the total sum of the FTD-CDR domain subscores that we called dementia severity (range 0–24). We separately included the score on the language domain that we called language severity (range 0–3). For the scoring, we used the “sum of the boxes” (SOB) method, which consists of the simple summing of each of the domain box scores (CDR-SOB). The CDR-SOB method demonstrated good reliability [49,50], and its use is supported as an index of severity for early cognitive impairment in therapeutic trials [51].

### 2.3. MRI Data Acquisition

Participants underwent MRI the same day of the baseline behavioral evaluation, except for nine patients who were scanned within 1 week of the structural brain imaging, three patients within 2 weeks and one patient 55 days after the behavioral assessment. Imaging data were acquired using a 3T Philips Achieva MRI scanner, equipped with a 32- channel head coil, to obtain axial MPRAGE T1—WIs (TR/TE = 8.1/3.76 ms) with a 224 × 224 matrix, FOV of 212 × 212 mm and 150 slices of 1 mm thickness. The T1-high resolution images were automatically segmented in a public web-based service for multi-contrast imaging segmentation and quantification, MRICloud (www.MRICloud.org, accessed on 15 December 2021) [52]. This process involves orientation and homogeneity correction; two-level brain segmentation (skull stripping) [53], then whole-brain image mapping based on a sequence of linear algorithms and large deformation diffeomorphic metric mapping (LDDMM) [54,55]. Forty-five JHU adult atlases (version 9) were used to generate 289 structural definitions and their respective volumes (in mm^3^) [56,57,58].

### 2.4. Statistical Analyses

#### 2.4.1. Demographic Differences between Variants

Fisher’s exact test for categorical variables (sex) and one-way ANOVAs were applied to compare the three PPA variant subgroups’ differences in demographic (age and education) and clinical features (years post onset, dementia severity, and language severity scores). The alpha level to determine significance was set at *p* < 0.05.

#### 2.4.2. Behavioral Differences in Letter and Semantic Fluency between PPA Variants

Letter and semantic fluency measures were converted into percentages of correct responses based on norms obtained from healthy participants, in order to easily compare performance in the two tasks [1]. 

To test for variant differences in performance between the two fluency tasks, we performed repeated-measures analyses of variance (ANOVAs) using a 2 × 2 design (variant × task), with the variant as the between-subjects factor and fluency task scores as the within-subjects factor. A Bonferroni post-hoc test was used for pairwise comparisons.

#### 2.4.3. Predictive Factors of Letter and Category Fluency

We a priori selected regions-of-interest (ROIs) based on areas involved in previous verbal fluency studies in PPA: bilateral pars opercularis (OpIFG), pars orbitalis (OrIFG), pars triangularis (TrIFG) of the inferior frontal gyrus [14,59], supramarginal gyrus (SMG) [21,59], anterior temporal pole (ATP) [60,61,62], middle temporal gyrus (MTG) [60], inferior temporal gyrus (ITG) [14,59,63], fusiform gyrus (FG) [59], dorsolateral prefrontal cortex (DLPFC), ventrolateral prefrontal cortex (VLPFC) [14,59], superior temporal gyrus (STG) [14,59,63], and angular gyrus (AG) [21,59]. 

All analyses were performed in native space; brain volumes for each ROI were normalized by the cerebral volume to control for brain size (calculated by adding the volumes of ROIs representing total brain tissue without myelencephalon and CSF). To control for individual brain atrophy, we calculated the ratio between the cerebral volume and the intracranial volume, the “Ratio ICV”, calculated by adding CSF to the cerebral volume [64]. We added Ratio ICV as a predictor in the regression model, according to recommendation to include initial brain tissue for determining brain–behavior relationships [65].

We performed a forward stepwise multiple regression model for each VF task using the cross-validated R^2^ as follows. In both models (one for letter and one for semantic fluency), the verbal fluency scores for each participant were entered as the dependent variable, whereas the gray matter volumes of the language areas (a priori ROIs), the overall atrophy (Ratio ICV), the clinical factors (years post-onset, dementia severity, and language severity scores), and the demographic factors (age, gender, and education) were entered as predictors to determine which factor explained most variance in patients’ performance on letter and semantic fluency. 

For each dependent variable, the first step of the stepwise regression starts from the model with no predictor areas, say model_0_, and finds the predictor, say Pred_model 0__→1_, that, when included into the model, gives the largest increase Δ(R^2^)_model 0__→1_, in the cross-validated R^2^, than if any other predictor were included instead. If this Δ(R^2^)_model 0__→1_ is positive, then we include that Pred_model 0__→1_ in the more accurate new model, say model_1_. Each next step continues similarly, to find if there is a predictor, among those not yet included in the model, that would produce a largest and positive increase in R^2^. Within steps, no problem with collinearity was detected as there was a definite choice of the predictor that increased R^2^ the most. Across steps, an implication of using the increase in cross-validated R^2^ is that it is a reliable indicator of the relative importance of the predictors in a model. 

Statistical level of significance was calculated in a stepwise fashion for each predictor and set at *p* < 0.05. All analyses were performed in R 3.3.2 software. 

## 3. Results

### 3.1. Demographic Differences between Variants

The three PPA variants did not significantly differ with regard to sex, age, education, symptoms duration (years post-onset) and severity (language and dementia). Means and standard deviations of demographic and clinical features for the 35 PPA patients and each of the three variants are reported in Table 1.

### 3.2. Behavioral Differences in Letter and Semantic Fluency between PPA Variants 

The ANOVA (variant x task) did not reveal any significant differences between tasks (F(2,35) = 1.719, *p* = 0.199), but showed an interaction between task and variant (F(2,35) = 13.952, *p* < 0.01). Post-hoc analyses revealed that participants with nfvPPA were significantly more impaired in letter compared to semantic fluency (*p* = 0.001); svPPA were significantly more impaired in semantic compared to letter fluency (*p* = 0.002); and LvPPA were equivalently impaired in letter and semantic fluency (see Figure 1). Means and standard deviations of accuracy scores in percentage for the letter and semantic fluency tasks per variant are reported in Table 2.

### 3.3. Predictors of Letter Fluency

The stepwise multiple regression model explained overall 55% of the variance of letter fluency performance. The most significant predictors were severity of language and the volumes of the left STG and of the right DLPFC. In descending order of total variance explained by the model, language severity accounted for 38% of the variance (with a negative regression coefficient), the left STG for an additional 12% (with a negative regression coefficient), and the right DLPFC accounted for another 5% (with a positive regression coefficient) (see Table 3 for *p* values).

The negative association between letter fluency and language severity is intuitively explained by the fact that high severity scores correspond to a worse performance (severity higher scores = more severe; letter fluency higher scores = less severe). In addition, the inverse correlation between the left STG and letter fluency means that poorer phonemic fluency is associated with greater left STG volume. The positive correlation with right DLPFC indicates that a larger volume in the right DLPFC corresponds to a more preserved phonemic fluency performance. 

We then added the PPA variant as a new independent variable with the other predictors in the regression model (for methods, see also [45]). The results remained mostly the same: language severity was the most significant predictor (its R-squared decreased to 20%, negative regression coefficient), the right DLPFC was the second most important predictor (its R-squared increased to 13%, positive regression coefficient) and the left STG was no longer a significant predictor, indicating its contribution may have been related to variants.

### 3.4. Predictors of Semantic Fluency

The stepwise multiple regression model explained overall 54% of the variance of semantic fluency performance. The most significant predictors were dementia severity and gray matter volume in the left ITG. In descending order of total variance explained by the model, dementia severity accounted for 47% with a negative regression coefficient since high severity scores correspond to low performance (severity higher scores = more severe; semantic fluency higher scores = less severe) and volume in the left ITG for an additional 7% (with a positive regression coefficient) (see Table 4 for *p* values).

We then added the PPA variant as a new independent variable with the other predictors in the regression model (for methods, see also [45]). The results for dementia severity remained the same: dementia severity was the most significant predictor with a similar contribution (its R-squared decreased slightly to 44%, negative regression coefficient). In addition, a related factor, overall atrophy, was found to significantly predict semantic fluency (R-squared = 7%). Finally, amongst the three variants, only the variants of lvPPA and svPPA contributed to semantic fluency (with a small but significant contribution, R-squared was 4%, negative regression coefficient for nfvPPA). The effect of gray matter volume in the left ITG was no longer significant, indicating that either LvPPA or svPPA or both contributed to atrophy in the left ITG.

## 4. Discussion

In the present study we aimed to determine: (1) the neuropsychological profiles of three variants of PPA on letter and semantic fluency, and (2) the neural, clinical and demographic predictors of letter and semantic fluency in PPA. The goal was to assess whether previous claims from the post-stroke literature—namely that letter fluency involves more frontal areas, but semantic fluency involves more temporal areas—apply to neurodegenerative conditions that involve language, such as PPA, while also controlling for symptom severity. Furthermore, we included, for the first time, patients with lvPPA, a variant with prevalent AD pathology [44]. We conducted stepwise multiple regressions for each fluency task and obtained models explaining 55% and 54% of the total variance in letter and semantic fluency, respectively. We confirmed the hypothesis that severity predicted performance of both letter and semantic fluency. Interestingly, language severity explained 38% of variance in letter fluency, whereas overall dementia severity explained 47% of variance in semantic fluency. When severity was controlled for, letter fluency performance was predicted by the volume of the left STG and right DLPFC, whereas semantic fluency performance was predicted by the volume of left ITG. We discuss these findings in light of the existing literature.

The finding that the overall dementia severity score predicted semantic fluency performance, but language severity predicted letter fluency performance is novel and noteworthy. This finding suggests that the access to conceptual knowledge (needed for the semantic fluency task) might be influenced by multidimensional degeneration involving different cognitive aspects of the disease, including semantic memory and global world knowledge. This multidimensional level of severity can be better assessed by the overall FTLD-CDR clinical assessment. Semantic processing relies on a large multidimensional network reflecting not only language deficits but also impaired memory and access to semantic knowledge. This important conclusion is further supported by the finding that overall (widespread) atrophy was also a predictor of semantic fluency independent of the variants. 

### 4.1. Behavioral Differences in Letter and Semantic Fluency between PPA Variants

In the present study, we replicated previous findings that: (1) patients with nfvPPA show a greater impairment in letter compared to semantic fluency; (2) patients with svPPA show a greater impairment in semantic compared to phonological fluency [30,43,66,67], and lvPPA show similar impairments in both letter and semantic fluency [27]. Our results in nfvPPA align with most previous studies showing impairments in both types of verbal fluency in this group but more impaired in letter than semantic fluency [28,30,43,67] except for the Libon and colleagues’ study [29]. The discrepancy between our results and the Libon and colleagues study [29] who found that there was no difference between letter and semantic fluency in nfvPPA may stem from two factors: (1) the participants with nfvPPA may have been more severe in Libon et al.’s study than in the present study (despite the similar illness duration) and therefore both letter and semantic fluency may have been impaired, and (2) the lvPPA was not included in all previous studies (e.g., [29,66]) or was considered together with the nfvPPA as belonging in the same “non-fluent” category [30,67]. The present study addressed both issues: (1) we included severity of symptoms as a predictor to be able to evaluate the contribution of each area independent of severity, and (2) we included lvPPA as a separate category according to consensus criteria. As mentioned above, we indeed found that severity plays a major role in explaining both letter and semantic fluency.

In our analyses, in addition to severity ratings, we included other clinical and demographic factors that were previously found to have an effect on language and fluency tasks: years post-onset, age, sex and education. Effects of age and education are commonly found in verbal fluency studies of healthy controls (e.g., [4,68]). Although these studies look at the difference in search strategies (clustering and switching), which was beyond the scope of the present study, age and education are usually good predictors of word retrieval strategies but sex is not [4,68]. In our previous volumetric analyses, we reported the role of language severity and education as significant predictors in action and object naming performance in PPA patients [45]. In the present study, however, we did not find any effects of education on letter or semantic fluency. We did not find an effect of sex, as Rogalski and colleagues did, especially in decline [43]. In their study, patients with PPA (without considering a differentiation by variant) and AD showed an effect of sex, i.e., women performed significantly worse compared to men on both letter and semantic fluency tasks and showed a more severe and rapid decline than men. Larger studies with the power to investigate the variant and sex interaction are needed to verify the role of sex in PPA. A possible explanation why we did not replicate this effect could be related to the fact that Rogalski and colleagues [43] found a sex effect in the overall Clinical Dementia Rating (CDR), whereas we used the FTLD-CDR (the battery used in our study) that includes a language subtest. It could be that although women decline more rapidly in cognition and are thus over-represented in AD, maybe they do not decline as much in language. 

### 4.2. Predictors of Letter Fluency

Imaging and lesion studies have already demonstrated the association between poor letter fluency and atrophy in frontal areas in both post-stroke aphasia and PPA. However, the role of each hemisphere has not drawn attention. Right lateralized frontal areas have been involved in letter rather than semantic fluency in AD and mild cognitive impairment (MCI) [69], as well as in healthy controls [64,70]. These results are also in line with the post-stroke, focal lesion literature, in which left frontal lesions correlate with impairments in letter fluency but temporal lesions do not [3,6,71]. Indeed, Henry and Crawford [3] found letter fluency to be more sensitive to left, as opposed to right, cortical focal lesions. By contrast, Laisney and collaborators [32] reported the involvement of right frontal regions with letter fluency in a group of fronto-temporal dementia patients. We would have expected age to be a significant predictor for letter fluency had we included patients from other age groups had we had a great variance.

The involvement of the right DLPFC has been explained as either (1) an effect caused by a reduction of the hemispheric asymmetry in late life [69] or (2) an effect of domain-general monitoring demands in working memor [70]. Our findings that letter fluency correlated positively with the volume of the right DLPFC, independently of language severity, could be due to either explanation. With regard to age-related hemispheric asymmetry, we do not have data to support an involvement of the right DLPFC in a healthy aging cohort. In our cohort, the variance in age was very low, i.e., patients were of the same age group (most above 65), and the variants did not differ in age. We would have expected age to be a significant predictor for letter fluency had we included patients from other age groups had we had a great variance. With regard to domain-general monitoring demands during letter fluency, the influence of the right DLPFC in letter fluency is confirmed by lesions [72,73] and imaging studies [74]. For example, Ramier and Hecaen confirmed the role of frontal regions on the verbal phonological searching responsible for the processing of “initiation of an action” in letter fluency, and they underlined the role of frontal right lateralized regions in the verbal domain of this “action initiation”. Zangwill [75] had also already pointed out that verbal fluency deficiency of the right frontal lesions is the result of a “loss of spontaneity and a difficulty in finding the appropriate words”.

Notably, the volume of the left STG was also negatively correlated with performance in letter fluency, i.e., larger volume (less atrophy) in the left STG corresponded to worse performance in letter fluency. We would like to speculate that this unexpected finding could be caused by the atrophy pattern of particular variant(s), because when we introduced the variant as a predictor, the left STG was no longer significant as a predictor of letter fluency. However, in absence of a control group to whom the volumes of the left STG could be compared, we cannot address this issue definitively.

With regard to the PPA literature, there is a finding in the Libon et al. study [29] that does not align well with the present results, or other results on neural substrates of letter fluency. Libon and colleagues found that atrophy in the left temporal regions correlated with letter fluency in svPPA and nfvPPA. It is important to consider that in the Libon et al. study (2009) the non-fluent group presented with a worse performance compared to the semantic group, both for letter and semantic fluency. Therefore, the non-fluent patients in the Libon et al. study (2009) might have been more progressed (higher clinical severity) compared to our nfvPPA patients, who were more impaired in letter than semantic fluency. In the present study, we controlled for this factor by including both language severity and overall dementia severity as predictors and were able to show that language severity was the most significant predictor of letter fluency. We also showed that the right DLPFC was a significant predictor of letter fluency independently from language severity or variants. The small contribution of involvement of the right DLPFC in letter fluency indicates that even a small change in the right hemisphere is detrimental enough to cause greater impairment than left atrophy.

### 4.3. Predictors of Semantic Fluency

It has been suggested that both verbal fluency tasks require frontally mediated executive retrieval mechanisms [3]. However, only semantic fluency requires additional access to more widely distributed semantic stores in the temporal cortex to search for exemplars of particular semantic categories. Semantic fluency seems to rely on both switching and clustering as phonological fluency does [4], and in addition, on accessing lexical/semantic representations. Indeed, according to focal lesions studies [3] and a metabolic study with semantic dementia patients [32], semantic fluency seems to rely on both frontal and temporal lobes. Our present findings for the neural substrates of semantic fluency in the left ITG, independent of severity, are in line with previous imaging studies in PPA patients [76], post-stroke aphasia [6], svPPA [29], and AD [69,77], as well as healthy controls [6,61]. Many studies, including some from our group, have shown the imperative role of the left ITG for lexical-semantic processing in lexical retrieval tasks such as naming and spelling [78,79,80].

However, in the Libon study (2009), semantic fluency in nfvPPA correlated with the left STG, as well as the right DLPFC, volume [29]. In that study, the nfvPPA patients performed worse than those with svPPA in the semantic fluency task. As this difference may have resulted from differences in severity, the present study controlled for language and dementia severity. We demonstrated that independently from severity, the volume of left temporal areas, particularly the left ITG, is an important predictor for semantic fluency. 

When a variant was entered in the regression, we found that only LvPPA and svPPA contributed significantly to semantic fluency (4% of the variance was explained by these two variants independently from the 44% of the variance explained by dementia severity). As noted in the beginning of the discussion, the association of semantic fluency with dementia severity demonstrates that impairment in semantic fluency reflects an impairment in global cognition.

### 4.4. Limitations

The main limitation of the present study is its sample size. Given the expected differences in verbal fluency performance and atrophy patterns across the different PPA variants, it would be ideal to examine areas predicting performance in letter and semantic fluency for each PPA variant separately (therefore running a stepwise model for the performance of each single variant). We believe that future research will focus on this type of analysis separately for each PPA variant. In conclusion, the present findings suggest that the letter and semantic fluency tasks can distinguish between PPA variants. Their brevity and the richness of the data acquired in these tasks make them particularly appropriate for quick evaluations in clinical settings.

## Figures and Tables

**Figure 1 brainsci-12-00001-f001:**
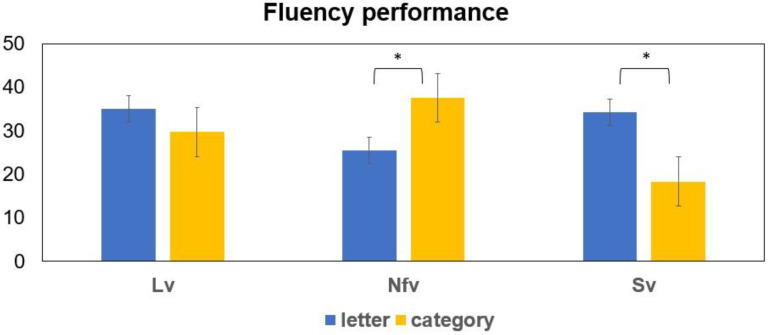
Behavioral scores of letter fluency and category fluency (accuracy is shown in percentage with standard error for both tasks) for the PPA variants (Lv = logopenic variant; Nfv = nonfluent variant; Sv = semantic variant), in the two fluency tasks (letter and semantic fluency). * *p* < 0.01.

**Table 1 brainsci-12-00001-t001:** Demographic information for all PPA patients and variants. F = female; yrs = years; Lv = Logopenic variant of PPA; Nfv = nonfluent variant of PPA; Sv = semantic variant of PPA; FTLD-CDR = fronto temporal lobar degeneration-specific Clinical Dementia Rating.

	TOT (N = 35)	Lv (N = 10)	Nfv (N = 17)	Sv (N = 8)	*p*-Values
Demographic	Mean (SD)	Mean (SD)	Mean (SD)	Mean (SD)	
Age	67.74 (7.6)	66.8 (9.7)	68 (7.6)	68.37 (4.8)	0.899
Gender	16F	6F	6F	4F	0.5
Education (yrs)	16.38 (2.3)	16.2 (2.5)	16.7 (2.3)	15.93 (2.4)	0.734
Onset (yrs)	4.34 (2.8)	4.69 (3.3)	3.39 (2.1)	5.9 (3.1)	0.112
Language score (FTDL-CDR 0–3)	1.77 (0.8)	1.8 (1)	1.58 (0.8)	2.12 (0.8)	0.377
Total Severity (FTDL-CDR 0–24)	6.78 (5.4)	8.25 (5.2)	5.38 (4.8)	7.93 (6.6)	0.34

**Table 2 brainsci-12-00001-t002:** Behavioral scores of letter fluency and category fluency (in percentage of correct responses with standard deviations in parentheses for both tasks) for all of the PPA patients and variants. Task differences: ^a^ = Nfv significantly impaired in letter compared to category fluency performance; ^b^ = Sv significantly impaired in semantic compared to letter fluency performance (*p* ≤ 0.05 with Bonferroni Tukey post-hoc corrections).

	Lv (N = 10)	Nfv (N = 17)	Sv (N = 8)
Tasks			
Letter fluency (F,A,S)	35 (23.46)	25.49 (15) ^a^	34.16 (23.78)
Category fluency (animals, fruits, vegetables)	29.66 (20.42)	37.54 (21.11)	18.33 (9.55) ^b^

**Table 3 brainsci-12-00001-t003:** Significant predictors of letter fluency: stepwise regression with demographic, clinical and ROIs as predictors. Severity L = language severity score at the fronto-temporal dementia scale; R DLPFC = right dorsolateral prefrontal cortex; L STG = left superior temporal gyrus. The added R-squared from the regression model refers to the additional variance explained by including the given variable.

Variable	B (SE)	β	*p*-Value	Model R²	“Added” R²
Severity L	−16.211 (2.877)	−5.635	<0.001	0.38	
L STG	−3515.876 (1527.765)	−2.301	0.028	0.50	0.12
R DLPFC	4253.253 (1850.83)	2.298	0.028	0.55	0.5

**Table 4 brainsci-12-00001-t004:** Significant predictors of category fluency. Stepwise regression with clinical, demographic and ROIs as predictors. Severity T = total severity score at the fronto-temporal dementia scale; L ITG = left inferior temporal gyrus. The added R-squared from the regression model refers to the additional variance explained by including the given variable.

Variable	B (SE)	β	*p*-Value		Model R²		“Added” R²
Severity T	−2.4277 (0.4232)	−5.737		<0.001		0.47	
L ITG	3315.3047 (1277.38)	2.595		0.014		0.54	0.07

## Data Availability

Data are available upon request from the corresponding author.

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
