# Peer review of "Neural Correlates of Letter and Semantic Fluency in Primary Progressive Aphasia"

_brainsci, 2021, doi:10.3390/brainsci12010001_

Round 1

Reviewer 1 Report

The authors contribute to a clinically and theoretically interesting topic by providing a paper of potential relevance in the literature. The methodology is correct and meets the standards. The methods section shows all the necessary information to be replicated and the statistical analysis section is clear, showing all the statistical tests used and a proper interpretation of the results.

The conclusions are in accordance with the described methods and results, avoiding causal inferences and showing statistical and substantial significance supported by appropriate theoretical frameworks.

I only have a few comments intended to improve clarity and readability

Introduction

The introduction is very complete and well-written. I only have requests for additional details or clarifications

  • P1, L42: the duration of VF tasks is not always 1 min.
  • P2, L53: I suggest adding language to the sentence: … and other cognitive functions related to memory and language (available semantic and lexical knowledge from which to identify relevant items) and …
  • P3, L102: According to Gorno-Tempini et al. (2011), nfvPPA can also present the characteristics of apraxia of speech, rather than agrammatism.
  • P3, L108: the reverse concreteness effect remains controversial in svPPA (see for example Cerebral Cortex, 21 (9), 2011.
  • P3, L111: The impairment in lvPPA affects the repetition of long sentences due to phonological short-term memory impairment.
  • P3, L133-142: The authors should reserve the term VF to VF tasks only to avoid confusion. Analyses in picture description, spontaneous speech, … rather refer to speech fluency
  • P3, L144-145: Libon et al. administered VF tasks to patients with svPPA, nfvPPA, and behavioral/dysexecutive disorder (not bvPPA).

Methods

  • P4, L203: incomplete sentence
  • Materials and procedures: please provide the scoring method used in VF tasks

Results

  • Add the results (p-values) of the statistical analyses on demographic variables in Table 1
  • P7, L338-343: the paragraph is confusing (incomplete sentences)

Discussion

  • P8, L367: please add the reference for the Libon et al. study and explain the nature of the discrepancy with this study
  • L367-376: the paragraph is confusing. Its introduction addresses the discrepancy regarding the results on nvfPPA but the paragraph addresses differences in the inclusion/exclusion of participants with lvPPA. Please clarify.

Author Response

We thank Reviewer 1 for the useful comments.

Reviewer 2 Report

Summary

The manuscript reports the results of a study investigating the predictors of performance on letter and category fluency tests in a group of 35 individuals with Primary Progressive Aphasia (PPA). In the study, neural predictors (i.e., volume in selected regions-of-interest (ROIs)) were entered along with demographic and behavioral (i.e., severity of language deficits and overall dementia severity) were entered in multiple regression analyses, with the aim to identify the best predictors of performance on both tasks. Results showed that letter fluency was best predicted by a combination of volume in the left superior temporal gyrus (STG) and in the right dorsolateral prefrontal cortex (DLPFC), as well as by severity of language deficits. Category fluency, on the other hand, was best predicted by volume in the left inferior temporal gyrus (ITG), and by overall dementia severity. The introduction of PPA variant in both analyses considerably changed the results, as volume in the STG was no longer a significant predictor of letter fluency accuracy, and volume in the left ITG was no longer a predictor of category fluency accuracy. Results are discussed as supporting the extant literature on the neural basis of letter and category fluency tasks, which links letter fluency to damage to frontal, and category fluency to temporal, areas.

Overall impression

The study addresses two important issues, which relate to 1) the neural mechanisms underlying letter and category fluency and 2) the clinical relevance of administering verbal fluency tasks – which are known to engaged non-linguistic (executive) functions – as part of the neuropsychological evaluation of individuals with PPA.

Although the sample size and the statistical analyses are appropriate, the description of both neuroimaging and statistical methods could be improved; namely, more details should be provided regarding the interactions/correlations between predictors entered in the regression models, as these may affect the parameter estimates. In addition, the motivation and overall scope of the study could be stated more clearly. The major issue with the manuscript lies in the Discussion, where some of the interpretations provided are not convincing and some findings that are potentially interesting are not discussed. My comments are below.

Major comments:

  • The rationale of the study could be stated more clearly. On Pages 3-4, some of the limitations of the previous literature investigating verbal fluency are discussed; however, although it can be inferred that the intent of the study is to overcome such limitations, the study aims are not explicitly stated and the motivation of the study seems lacking. On the other hand, the aim of the study is clearly stated in the Discussion (page 7, lines 347-350). Therefore, the text on pages 3-4 could be condensed and re-organized to better provide a motivation for the present study, and the aim should be clearly stated at that point.
  • On Page 6 (lines 285-289), the authors provide a list of the variables included in the regression analyses, but do not explain how variables were entered in the analyses and if/how the relationship between the predictors was assessed. Namely, they manuscript should indicate a) if continuous variables were mean-centered prior to be included in the regression analyses, b) if collinearity between predictors was checked, and how collinearity between predictors (if any) was handled, c) if step-wise forward or backward regression were run, and d) how the decision on whether to keep or discard a predictor was made (e.g., model comparison; p-value, R-squared value, etc.)
  • The authors selected two measures of severity to include in their regression models: 1) the total FTD-CDR score (which is considered as an index of the severity of dementia, as it encompasses several domains of cognition and behavior), and 2) the score obtained on the language subscale of the FDT-CDR (index of severity of language deficits). The results indicate that, while the first measure (overall dementia severity) is a predictor of performance on category fluency tasks, the second is a predictor of performance on letter fluency tasks. This seems a very interesting result, which is not addressed in the Discussion, where dementia severity and language severity seem to be treated as virtually identical (see, for example, Page 7, line 355, and Page 8, lines 376-377).
  • In the Discussion of the dissociation between performance on letter fluency and category fluency tasks (page 7, line 360), could a mention be made on healthy participants’ behavior? Although I understand that data from healthy participants was not collected as part of this study, are there any data from healthy participants available in the literature? This would help determine if one task is harder than the other in the normal population.
  • The interpretation of the regression results for the letter fluency task is somewhat incomplete and not justified. Results show that – when diagnosis of nfvPPA was introduced as a predictor in the analyses – the negative relationship between volume in the left STG and performance on the letter fluency task was no longer significant. This result is interpreted as supporting the idea that letter fluency relies on frontal regions (where participants with nfvPPA typically show atrophy) and not on temporal regions (where participants with nfvPPA typically do not show atrophy). However, this conclusion is not sufficiently supported, in the absence of 1) evidence that the nfvPPA show no atrophy (compared to healthy participants) in the STG, and/or 2) evidence that the nfvPPA show greater gray matter volume in the STG than both the other variants. In addition, this conclusion is partly at odds with the lack of association between volume in left frontal regions (IFG or DLPFC) and performance on the letter fluency task. A similar argument is made when discussing the results of the regressions on category fluency (Page 7, lines 341-343, and in the Discussion), which should be substantiated by statistical evidence that atrophy in the left ITG was lesser (or none) in the nfvPPA group, when compared to the other variants. Moreover, the interpretation provided on Page 7 (line 341-342: “they absorbed the ITG contribution”) should be better explained.
  • Along the same lines, in discussing the findings from the category fluency task (Pag 9, lines 436-438), the authors should comment on the fact that volume in the left ITG only predicted a very small proportion of variance, and - most importantly – that its contribution disappeared when the RatioICV was entered in the regression. The latter suggests that semantic category fluency is predominantly explained by widespread atrophy (and consequent impairment of global cognition). This should be addressed.
  • The interpretation of the role of the right DLPFC in letter fluency tasks should be amended or clarified (Page 8, lines 404-406). If involvement of the right DLPFC is due to reduction of hemispheric asymmetry in late life, then, shouldn't we expect age to have an effect? Also, with respect to the other explanation (i.e., effect of domain-general monitoring demands in working memory), shouldn't we find that - when introducing dementia severity - the effect of the right DLPFC volume is no longer significant?

Minor comments

  • Page 2, lines 62-64: A citation (or further explanation) should be provided for the fact that LF is more likely to require switching due to reliance on orthographic search; this is not intuitive
  • Page 2, line 71: “focal lesions” next to the reference should be deleted; in addition, the following sentence starting with “Either, both…” is not entirely grammatical and should be clarified.
  • Page 3, lines 133-138: It seems the authors are considering verbal fluency as similar to spontaneous speech. Although the two tasks undoubtedly entail some of the same processes, spontaneous speech is a naturalistic task that provides many more qualitative and quantitative measures of language performance and that assesses more levels of processing than verbal fluency tasks. Since I am unsure what is the point that the authors are trying to make here, my suggestion is to leave this entire paragraph out.
  • Page 4, line 161: “Another possible reason… AND the previous literature”
  • Page 4, lines 179-180: “…other studies DID NOT confirm a gender effect EITHER on LF (Barry et al., 179 2008; Boone et al., 1999; Tombaugh et al., 1999) OR on CF (Lucas et al., 1998)…”
  • Page 4, lines 183-185: “Instead, education predicted both tasks (Crossley et al., 1997) or just LF performance (Ruff et al, 1997; Tombaugh et al,, 1999)” -> the sentence sounds ungrammatical
  • Page 4, lines 197-198: “Therefore, we added as predictors in the regression models the cerebral volumes associated with literature in fluency and PPA…” -> I think what the authors mean here is something along the lines: “Therefore, we included volume in regions of interests (selected accordingly to the previous literature on verbal fluency in PPA) as predictors in regression models…”
  • Page 5, line 222: “The score may range…” -> “Scores range…”
  • Page 5, line 245: the acronym LDDMM is not defined
  • Page 6, line 285: “volumes of the language areas (a prior ROIs)” -> were only gray matter images used? This is not clearly stated.
  • Page 6, line 308: “the left STG and the right DLPFC” -> “the volume of the left STG and of the right DLPFC?”
  • Page 6, line 310 (and elsewhere in the manuscript): “(with a negative correlation coefficient)” -> since the authors ran regressions, it would be more appropriate to talk about “regression coefficient”
  • Page 7, lines 315-318: This statement should be moved to the beginning of the next paragraph, as it seems to constitute the premise to the additional analysis they ran.
  • Page 7, lines 338-340: The sentence is not grammatically correct.
  • Page 7, line 340: “The left ITG disappeared” -> The effect of gray matter volume in the left ITG was no longer significant.
  • Page 9, lines 429-430: a reference should be provided here.

Author Response

We thank Reviewer 2 for the useful comments. They resulted in a much improved manuscript.

Please see response below.

Round 2

Reviewer 2 Report

The authors have addressed all of my comments and have delivered a manuscript that has significantly improved in clarity and readability. I especially appreciate the changes made to the Discussion, which has been considerably expanded to include possible explanations for some interesting findings that were reported in the results.

I only have a few minor comments:

  • Page 2 (lines 70-71): “…has been reported, since some studies have shown reduced letter fluency following left RATHER than right frontal lesions…”
  • Page 4 (line 187): …letter and SEMANTIC fluency…” (just to be consistent with the rest of the manuscript).
  • Page 5 (line 216): “…and demographic FACTORS…”
  • Page 5 (line 250): “…included the score ON the language domain that we called…”
  • Page 5 (line 258): “… except FOR 9 patients…”
  • Page 6 (lines 281-283): how was this calculated? Based on norms obtained from healthy participants on each task (Letter and Semantic fluency)? Or was the number of words produced on letter (or semantic) fluency divided by the total number of words recalled across letter and semantic fluency (i.e., the total raw score described on page 5)? If so, this would provide a measure of how better performance on one task is compared to the other, rather than a general measure of accuracy on the task.
  • Page 6 (line 290): “…verbal fluency studies IN PPA…”
  • Page 7 (line 358): “Therefore, the inverse correlation…” -> Perhaps “IN ADDITION, the inverse correlation…”?
  • Page 7 (line 365): “…R-squared decreased TO 20%…”
  • Page 8 (lines 374-375): “…were dementia severity and GRAY MATTER VOLUME IN the left ITG.”
  • Page 8 (line 377): “… and VOLUME IN the left ITG for an additional 7%...”
  • Page 8 (line 380): “…results FOR dementia severity…”
  • Page 8 (lines 384-387): the sentence “Finally, amongst the 3 variants […] to semantic fluency” seems contradictory (but the result is clearly stated in the Discussion, Page 11 (lines 561-562).
  • Page 8 (lines 417-418): “…reflecting not only language deficits but also IMPAIRED memory and access to semantic knowledge”.
  • Page 9 (line 431): “…except FOR the Libon and colleagues…”
  • Page 9 (line 436); “… may have of different degrees of severity […] may have been impaired”. Perhaps “more severe in the Libon et al.’s study than in the present study”?
  • Page 10 (line 491) and Page 11 (line 530): “…independently OF language severity…”
  • Page 11 (line 562): “…contributed significantly TO semantic fluency…”.

Author Response

We would like to thank the anonymous reviewer for the reading our manuscript so thoroughly. The comments have greatly improved the quality of our report.

We have addressed and corrected all the minor issues noted as well.